# Heterostructured α-Bi_2_O_3_/BiOCl Nanosheet for Photocatalytic Applications

**DOI:** 10.3390/nano12203631

**Published:** 2022-10-16

**Authors:** Daoguang Teng, Jie Qu, Peng Li, Peng Jin, Jie Zhang, Ying Zhang, Yijun Cao

**Affiliations:** 1School of Chemical Engineering and Zhongyuan Critical Metals Laboratory, Zhengzhou University, Zhengzhou 450001, China; 2School of Ecology and Environment, Zhengzhou University, Zhengzhou 450001, China

**Keywords:** Bi_2_O_3_, dechlorination, heterostructure, photocatalytic degradation, organic dye

## Abstract

Photocatalytic degradation of organic pollutants in wastewater is recognized as a promising technology. However, photocatalyst Bi_2_O_3_ responds to visible light and suffers from low quantum yield. In this study, the α-Bi_2_O_3_ was synthetized and used for removing Cl^−^ in acidic solutions to transform BiOCl. A heterostructured α-Bi_2_O_3_/BiOCl nanosheet can be fabricated by coupling Bi_2_O_3_ (narrow band gap) with layered BiOCl (rapid photoelectron transmission). During the degradation of Rhodamine B (RhB), the Bi_2_O_3_/BiOCl composite material presented excellent photocatalytic activity. Under visible light irradiation for 60 min, the Bi_2_O_3_/BiOCl photocatalyst delivered a superior removal rate of 99.9%, which was much higher than pristine Bi_2_O_3_ (36.0%) and BiOCl (74.4%). Radical quenching experiments and electron spin resonance spectra further confirmed the dominant effect of electron holes h^+^ and superoxide radical anions ·O_2_^−^ for the photodegradation process. This work develops a green strategy to synthesize a high-performance photocatalyst for organic dye degradation.

## 1. Introduction

Current industrial productions generate large amounts of pollutants. Hazardous organic contaminants that exist in industrial wastewater can endanger the whole ecosystem and human health [1]. Rhodamine B (RhB, C_28_H_31_ClN_2_O_3_) is one of the most commonly used cationic dyes applied in the textile, painting, medicine and cosmetic industries [2]. Due to its high solubility in aqueous solutions and organic solvents, RhB compound has strong irritation to the skin, eyes and respiratory tract, and can even have potential carcinogenic effects [3,4]. The common removal methods of RhB include adsorption, ion exchange, photocatalytic degradation and biological treatment [5,6,7,8]. Among these methods, photocatalytic degradation is recognized as an effective technology with a high removal rate [9].

It is critical to develop a highly efficient photocatalyst for the photocatalytic degradation process. TiO_2_ is a classical photocatalyst with a wide band gap (3.2 eV) that only allows it to be excited under ultraviolet (UV) light radiation (which only occupies ~5% of the solar light) [10,11]. In last decade, various visible light responding metal oxides/sulfides have attracted attention, such as WO_3_ (2.7 eV), Bi_2_O_3_ (2.6–2.8 eV), SrNbO_3_ (2.3 eV) and CdS (2.4 eV) [12,13,14,15]. Due to low toxicity, bismuth-based semiconductor Bi_2_O_3_ has emerged as an alternative candidate [16]. Generally, Bi_2_O_3_ has two common polymorphs: monoclinic α phase (~2.8 eV) and tetragonal β phase (~2.6 eV) [17]. Previous studies mainly focused on the β-Bi_2_O_3_. For example, Brezesinski et al. [18] used β-Bi_2_O_3_ to decompose RhB and the catalyst showed a degradation rate of ~80% within 2.5 h. However, the β-Bi_2_O_3_ is a thermodynamic metastable phase [19], and there were few reports about stable α-Bi_2_O_3_. Currently, the low quantum yield originating from the rapid recombination of photogenerated electron-hole pairs limits the practical application of α-Bi_2_O_3_ [20]. To improve photocatalytic performances of Bi_2_O_3_, heterostructure is an available approach [21]. The heterogeneous interface between different compositions in heterostructure-based material can modify electronic band structure [22,23]. Due to same element of Bi and the close band gap of ~3.5 eV, BiOCl is an ideal composition matched to Bi_2_O_3_ [24]. Furthermore, the layered structure of BiOCl can shorten photoelectron transmission distance and accelerate the separation of photogenerated electron-hole pairs [25,26]. Combing a narrow band gap with a distinctive nanostructure is an appealing strategy to simultaneously achieve visible light response and high quantum yield. The BiOCl is usually prepared through the reaction between Bi_2_O_3_ and HCl [27]. It is still a huge challenge to develop a green method for synthetizing a Bi_2_O_3_/BiOCl heterostructured photocatalyst.

Chloride ion (Cl^−^) is also a common inorganic pollutant in strongly acidic wastewater from the metallurgical industry [28]. The high Cl^−^ content could cause metal corrosion, soil salinization and human diseases. The conventional removal technique of Cl^−^ in acidic wastewater is to use Cu(0) and Cu(II) for generating CuCl precipitate [29]. Unfortunately, the low removal efficiency (<60%) would lead to a high concentration of residual Cl^−^ (>400 mg/L). In view of this, the Bi_2_O_3_ could be applied to capture Cl^−^ in acidic wastewater and be transformed to the BiOCl product [30].

Herein, the as-prepared α-Bi_2_O_3_ was used to remove high concentration Cl^−^ wastewater to generate the BiOCl. By coupling Bi_2_O_3_ with layered BiOCl, the α-Bi_2_O_3_/BiOCl composite material exhibited the nanosheet structure. The heterogeneous interface between Bi_2_O_3_ and BiOCl can serve as “heterojunction” to enhance the photocatalytic activity due to efficient charge separation and transfer across the interface [31,32]. In this case, the binary Bi_2_O_3_/BiOCl catalyst delivered a larger degradation rate of RhB under visible light irradiation than pristine Bi_2_O_3_ or BiOCl.

## 2. Experimental

### 2.1. Preparation of α-Bi_2_O_3_

All reagents were of analytical grade and purchased from Sinopharm Group Co. Ltd., Shanghai, China. On the basis of previous literature [19], 12.13 g of Bi(NO_3_)_3_·5H_2_O (25 mmol) was dissolved in 50 mL of deionized H_2_O and 4 mL of HNO_3_ (68 wt%) under continuous stirring of 300 RPM for 10 min. NH_3_·H_2_O (28 wt%) was added dropwise into the above solution until pH was ~7, and then the solution was stirred under ultrasonic radiation at 80 °C for 10 h. After filtration and drying, the residue was calcined at 500 °C for 2 h to obtain the yellow α-Bi_2_O_3_ product.

### 2.2. Preparation of BiOCl and B_2_O_3_/BiOCl from Adsorbed Cl^−^

BiOCl was synthetized by using Bi_2_O_3_ to adsorb Cl^−^ in wastewater. Typically, 1.64 g of as-prepared Bi_2_O_3_ powder (3.52 mmol) was dissolved in 100 mL NaCl aqueous solution (initial Cl^−^ concentration of 1000, 2000, 2500, 3000, or 4000 mg/L) under stirring at 300 RPM [33]. Using HNO_3_ (68%) as the pH modifier, the Cl^−^ concentrations at different contact times (0–180 min) were determined by ion chromatograph. After thorough washing and drying, the yellow-green BiOCl product was obtained (Equation (1)) [30]. The effects of pH (1–5), molar ratio of Bi^3+^:Cl^−^ (2.5:1, 1.25:1, 1:1, 0.833:1, and 0.625:1) and initial Cl^−^ concentration (1000–4000 mg/L) on the removal efficiency of Cl^−^ were investigated. The preparation of the grey Bi_2_O_3_/BiOCl was the same as BiOCl, but with 100 mL NaCl solution replaced with the 40 mL.
Bi_2_O_3_ + 2H^+^ + 2Cl^−^ = 2BiOCl + H_2_O(1)

### 2.3. Characterizations

Bruker AXS D8 ADVANCE diffractometer (Cu Kα source) (Bremen, Germany) was employed to characterize the X-ray diffraction (XRD) patterns. Surface functional groups were tested using a Fourier transform infrared spectroscopy (FTIR) analyzer (Nicolet iS10, Thermo Fisher Scientific, Waltham, MA, USA) mixed with KBr. Element compositions were characterized by an ESCA LAB MK-II X-ray photoelectron spectrometer (XPS, VG Scientific, Uppsala, Sweden). Micromorphologies were investigated by the field emission scanning electron microscopy (FESEM, SU8010, 30 kV, Hitachi, Ibaraki, Japan) with an energy-dispersive spectrometer (EDS) and transmission electron microscopy (TEM, JEM-2100F, 200 kV, JEOL, Tokyo, Japan). Specific surface areas were analyzed at 77 K using the Brunauer–Emmett–Teller (BET) method via the Micromeritics analyzer (Autosotrb-IQ2-MP-XR, Norcross, GA, USA). Thermal behavior was analyzed by a thermogravimetric (TG) analyzer (NETZSCH, STA449F3, Selb, Germany) with a heating rate of 5 °C/min in a nitrogen atmosphere. The UV–vis diffuse reflectance spectra (UV–vis DRS) were recorded on a UV-3600 PLUS spectrophotometer (Shimadzu, Kyoto, Japan). Electron spin resonance (ESR) spectra were obtained by Bruker EMXPLUS (Bremen, Germany). Intermediates of photocatalysis were detected by liquid chromatography-mass spectrometer (LC–MS, Ultimate 3000 UHPLC-Q Exactive, Thermo Fisher Scientific, Waltham, MA, USA) equipped with an elaectrospray ionization (ESI) positive ion mode.

### 2.4. Photocatalytic Degradation Measurement

The photocatalytic degradation performances of the materials were evaluated by using RhB solution. First, 20 mg Bi_2_O_3_ or Bi_2_O_3_/BiOCl was added into 100 mL RhB solution (20 mg/L) under stirring of 300 RPM at room temperature for 30 min in the dark, and then a 300 W Xe lamp (Asahi Spectra, MAX-303, Tokyo, Japan) was used for photocatalytic irradiation. The control experiments of adsorption were conducted as the same as photocatalytic degradation, but without light radiation during the whole process. The RhB concentration in the solution was analyzed using the ultraviolet-visible (UV–Vis) spectrophotometer (Shimadzu, UV-3600 PIUS, Kyoto, Japan) at wavelength of ~550 nm.

Radical quenching experiments were conducted to detect the dominant active species for RhB degradation. Isopropanol (IPA, 10 mmol/L), ethylenediaminetetraacetic acid disodium salt (EDTA-2Na, 5 mmol/L) and 1,4-benzoquinone (BQ, 1 mmol/L) were employed to quench hydroxyl radical (·OH), hole (h^+^) and superoxide radical (·O_2_^−^), respectively [34]. All the experiments were conducted under the same conditions as in the photocatalytic degradation test except for the addition of a separate scavenger.

Removal rate (*R_t_*) of pollutant was calculated using Equation (2).
(2)$$R_{t}\;=\;\frac{C_{0}\;-\;C_{t}}{C_{0}}\;\times \;100\%$$
where *C*_0_ and *C_t_* are the concentration of pollutant at contact time of 0 and *t*, respectively.

Reaction kinetic study was evaluated using the pseudo first-order model of Equation (3) [17].
(3)$$-\ln\left(C_{t}/C_{0}\right)\;=\;k\;\cdot \;t$$
where *k* is the apparent rate constant of equation.

## 3. Results and Discussion

### 3.1. Physical Properties of Materials

According to Appendix A, Bi_2_O_3_ presented excellent adsorption capacity of Cl^−^ in the acidic solution. Under the optimal conditions of pH = 1, initial Cl^−^ concentration = 2500 mg/L, Bi^3+^:Cl^−^ ratio = 1:1, contact time = 180 min and the removal rate of Cl^−^ could be reached to >95%, which suggests that there is a low concentration of residual Cl^−^ (<120 mg/L) [30]. The dechlorination products of BiOCl and Bi_2_O_3_/BiOCl were further used for photocatalysts.

The XRD patterns of Bi_2_O_3_, BiOCl and Bi_2_O_3_/BiOCl are illustrated in Figure 1a. For the Bi_2_O_3_, two sharp peaks located at around 27.5° and 33.3° could be attributed to the (120) and (200) planes of monoclinic α-Bi_2_O_3_ (JCPDS #72-0398), respectively [20]. Several diffraction peaks at approximately 11.9°, 25.8°, 32.5° and 33.4° corresponded to the (001), (101), (110) and (102) planes of tetragonal BiOCl (JCPDS #85-0861), respectively [35]. The Bi_2_O_3_/BiOCl composite displayed a hybrid pattern with characteristic peaks of α-Bi_2_O_3_ and BiOCl, which suggests that it maintained the original crystal structure of the two compositions. All materials showed sharp diffraction peaks, which indicated their good crystallinity [19].

Figure 1b presented the FTIR spectra of three materials. For the Bi_2_O_3_, two broad absorption peaks at ~3450 and ~1620 cm^−1^ could be classified as the stretching and bending vibrations of the O–H bond, respectively [20]. A peak located at ~850 cm^−1^ was attributed to the stretching vibration of Bi–O–Bi bond, and a sharp peak at ~520 cm^−1^ was assigned to the stretching vibration of Bi–O bond [36]. With regard to the BiOCl, the Bi–O–Bi peak disappeared and a new peak appeared at ~1380 cm^−1^ (stretching vibration of Bi–Cl bond) [37]. The Bi_2_O_3_/BiOCl simultaneously contained the Bi–O–Bi and Bi–Cl peaks, which indicated the successful preparation of such composite material.

The porosities of Bi_2_O_3_, BiOCl and Bi_2_O_3_/BiOCl were characterized by the liquid nitrogen adsorption–desorption isotherms in Figure 1c. All materials exhibited type IV isotherms with a hysteresis loop, which suggests the existence of mesopores (2–50 nm) [38]. Among the three samples, the Bi_2_O_3_/BiOCl expressed the largest N_2_ adsorption volume, which suggests its maximum specific surface area. Using the BET method, the surface area of Bi_2_O_3_/BiOCl was calculated as 11.2 m^2^/g, which was much higher than the 2.3 m^2^/g of Bi_2_O_3_ and 6.8 m^2^/g of BiOCl. In addition, Appendix A reveals the Barrett–Joyner–Halenda (BJH) pore size distributions of three samples [39]. Similarly, the Bi_2_O_3_/BiOCl composite delivered the largest pore volume with an average pore diameter of 20.7 nm (Appendix A). The large surface area and pore volume of Bi_2_O_3_/BiOCl can expose more active site, which is beneficial for subsequent adsorption and photocatalysis [15,40].

The TG (25–900 °C) analysis was used to investigate the thermal stability of Bi_2_O_3_/BiOCl. From Appendix A, the small weight loss on the TG curve below 700 °C could be assigned to the absorbed water. Until the temperature rose to 700 °C, an obvious decline started to appear at 700–800 °C, which can be attributed to the pyrolysis of the BiOCl composition [35]. The result revealed the excellent thermal stability of the Bi_2_O_3_/BiOCl material.

The XPS survey spectra of Bi_2_O_3_, BiOCl and Bi_2_O_3_/BiOCl is presented in Appendix A. For the Bi_2_O_3_, some obvious peaks at around 159, 285, 442 and 531 eV were attributed to the Bi 4f, C 1s, Bi 4d and O 1s, respectively [41]. Two new peaks appeared at 197 and 268 eV in BiOCl and Bi_2_O_3_/BiOCl spectra could correspond to the Cl 2p and Cl 2s. It confirmed the successful introduction of Cl^−^ [37]. For the high-resolution Bi 4f spectra in Figure 1d, all samples displayed two spin-orbit doublet peaks of Bi 4f_7/2_ (159.2 eV) and Bi 4f_5/2_ (164.5 eV). The above doublet peaks could be attributed to the presence of Bi^3+^ [38]. The O 1s spectra (Figure 1e) could be deconvoluted into two compositions. The peak at 529.8 eV was assigned to the lattice oxygen of Bi–O–Bi and the peaks at 531.3 eV originated from adsorbed oxygen in the water molecule (H–O–H) [27]. The Cl 2p high-resolution spectra (Figure 1f) also expressed two spin-orbit doublet peaks of Cl 2p_3/2_ (198.0 eV) and Cl 2p_1/2_ (199.5 eV) for the characteristic of Cl^−^ [39].

Figure 2 shows the SEM images of three samples. As observed in Figure 2a, the as-prepared Bi_2_O_3_ was stacked by interlaced nanorods with irregular shapes [19,42]. These irregular nanorods had the particle size of 100–300 nm (Figure 2b). After chlorination, the nanorods disappeared and there were only irregular nanoflakes for the BiOCl (Figure 2c,d) [27,43]. For the Bi_2_O_3_/BiOCl composite, Figure 2e displayed a distinctive structure different from both Bi_2_O_3_ and BiOCl. The Bi_2_O_3_/BiOCl was assembled by numerous staggered nanosheets with a diameter of 80–200 nm (Figure 2f). Such a distinctive nanosheet array structure of Bi_2_O_3_/BiOCl can devote a large surface area and provide more active sites for photocatalytic reactions [26,33]. The mapping images (Figure 2g) detected the uniform distribution of Bi, Cl and O elements on Bi_2_O_3_/BiOCl [37,44].

The TEM images were used to further reveal the microstructure of Bi_2_O_3_/BiOCl. From Figure 3a,b, the Bi_2_O_3_/BiOCl presented a thin nanosheet structure, which was consistent with the SEM results. The high resolution TEM (HR-TEM) in Figure 3c showed clear lattice fringes. These lattice fringes exhibited different arrangement directions with various crystalline domains, and could be attributed to Bi_2_O_3_ and BiOCl nanocrystallines [42]. For example, the lattice fringes with d-spacing of 0.33 nm were assigned to the (120) plane of Bi_2_O_3_, and the lattice fringes with d-spacing of 0.74 nm were assigned to the (001) plane of BiOCl (Figure 3d) [30,45]. A fast Fourier transform (FFT) image (inset in Figure 3d) illustrates regular lattice patterns for monoclinic crystal (Bi_2_O_3_) and tetragonal crystal (BiOCl), verifying two compositions of Bi_2_O_3_/BiOCl [30]. The heterogeneous interface of the material can modify electronic structure and improve the photocatalytic performances [46].

### 3.2. Photocatalytic Performances

UV–Vis DRS were conducted to characterize the optical absorption properties of Bi_2_O_3_, BiOCl and Bi_2_O_3_/BiOCl. As illustrated in Figure 4a, Bi_2_O_3_ and Bi_2_O_3_/BiOCl possessed strong absorption in the UV light region compared with that of BiOCl [47]. The absorption edges of Bi_2_O_3_, Bi_2_O_3_/BiOCl and BiOCl were 440, 380 and 360 nm, respectively. Band gap energy (*E*_g_) of the semiconductor photocatalyst can be calculated according to the Beer-Lambert law (Equation (S1)) based on the absorption edge [17,48]. The *E*_g_ values of Bi_2_O_3_ and BiOCl were calculated as 2.85 eV (Figure 4b) and 3.52 eV (Figure 4c). In addition, the positions of the valence band edge (*E*_VB_) and conduction band edge (*E*_CB_) for Bi_2_O_3_ and BiOCl were also estimated according to Equations (S2) and (S3) [17,49].

The optical properties of the materials are summarized in Appendix A and Figure 4d. Based on analysis above, the Bi_2_O_3_/BiOCl composite was a hybrid photocatalyst with two compositions. Compared to pristine Bi_2_O_3_ or BiOCl, these hybrid compositions can modify band structure and enhance the photocatalytic activity [41,45].

RhB was employed as the target pollutant to investigate the photocatalytic performances of the materials. Figure 5a showed the adsorption effect of three samples in darkness. The RhB can exist stably in the solution during the whole process without any photocatalyst (blank curve) [37]. Adding the materials, all three curves achieved the adsorption–desorption equilibrium at a contact time of 30 min. In addition, the Bi_2_O_3_/BiOCl exhibited the maximum adsorption capacity (minimum *C_t_*/*C*_0_ value). The reason could be attributed to the large surface area of Bi_2_O_3_/BiOCl [38].

The photodegradation behavior of RhB was displayed in Figure 5b. After the adsorption process in darkness for 30 min, there was no obvious degradation for RhB by using the Bi_2_O_3_. The Bi_2_O_3_/BiOCl composite presented significant degradation of its performance, with a total removal rate of ~99.9% at an irradiation time of 60 min (determined from the UV–Vis absorbance spectra of RhB in Figure 5c) [20]. By contrast, the total removal rates of Bi_2_O_3_ and BiOCl were only recorded as 36.0% and 74.4%, respectively. Accordingly, the removal rate of RhB increased as Bi_2_O_3_/BiOCl > BiOCl > Bi_2_O_3_ [41,42]. The surface area values of the catalysts exhibited the positive correlation with the removal rate of RhB, which suggests that the larger surface area can provide more active sites for photocatalytic reaction.

Figure 5d illustrated the pseudo first-order kinetic model of RhB photodegradation [17]. Using the linear fitting, the Bi_2_O_3_/BiOCl showed the steepest straight line with the largest slope value among three samples, which indicates its excellent photocatalytic activity [35,37]. The apparent rate constant *k* of kinetic equation was recorded in Figure 5e. As expected, the Bi_2_O_3_/BiOCl catalyst had the maximum *k* value of 0.1061 min^−1^. The *k* value was almost 100 times higher than that of Bi_2_O_3_ (0.0013 min^−1^).

Norfloxacin and tetracycline hydrochloride (TCHC) were used as targets to further estimate the photodegradation performances of Bi_2_O_3_/BiOCl. Similarly, after adsorption in darkness for 30 min, both the norfloxacin and TCHC expressed obvious degradation (Figure 5f). At irradiation time of 120 min, the total removal rates of norfloxacin and TCHC were recorded as 59% and 63%, respectively. The high removal rates of organic pollutants (e.g., RhB, norfloxacin and TCHC) demonstrated the availability of our Bi_2_O_3_/BiOCl photocatalyst [36,48].

The stability for the Bi_2_O_3_/BiOCl catalyst after 60 min photocatalytic degradation of RhB was evaluated. As shown in Figure 6a, the catalytic Bi_2_O_3_/BiOCl sample presented a similar XRD pattern to pristine Bi_2_O_3_/BiOCl, with significant characteristic peaks of monoclinic Bi_2_O_3_ and tetragonal BiOCl. Such results confirmed the good structure stability of Bi_2_O_3_/BiOCl material [49,50].

Figure 6b exhibited the high-resolution Bi 4f spectrum of Bi_2_O_3_/BiOCl after degradation testing. The sample showed two spin-orbit doublet peaks of Bi 4f_7/2_ (159.0 eV) and Bi 4f_5/2_ (164.3 eV) for the Bi^3+^. For the O 1s spectrum (Figure 6c), it included two deconvolutional peaks: lattice oxygen in Bi–O–Bi (529.6 eV) and adsorbed oxygen in the water molecule (H–O–H) (531.8 eV). Compared to the pristine Bi_2_O_3_/BiOCl, the peak corresponding to the O was still maintained [41,45]. A slight shift toward a lower binding energy (~0.3 eV) of the Bi–O peak was ascribed to the lower oxidation state of Bi atoms [43].

As observed in Figure 6d, the catalytic Bi_2_O_3_/BiOCl sample still displayed the nanosheet shape. The results further demonstrated the potential application opportunity of the Bi_2_O_3_/BiOCl photocatalyst [50].

Radical quenching experiments were performed to explore the potential photodegradation mechanism of RhB using Bi_2_O_3_/BiOCl. IPA, EDTA-2Na and BQ were employed as scavengers to quench hydroxyl radicals (·OH), electron holes (h^+^) and superoxide radical anions (·O_2_^−^), respectively [51]. From Figure 7a, adding the IPA into the system made almost no difference to the entire photocatalytic degradation rate. However, EDTA-2Na and BQ (especially EDTA-2Na) greatly inhibited Bi_2_O_3_/BiOCl from degrading RhB. In other words, ·OH did not play a role here while h^+^ and ·O_2_^−^ dominated the photocatalytic degradation of RhB by Bi_2_O_3_/BiOCl [39,52]. Using 5,5-dimethyl-1-pyrroline N-oxide (DMPO) as a capture agent, the ESR electron spin capture technology was performed to further detect the generation of ·OH and ·O_2_^−^. No ⋅OH signals appeared whether or not there was light (Appendix A). As for the ·O_2_^−^ active free radical, no signal was generated in the dark, while the obvious characteristic peak of the ·O_2_^−^ spectrum was observed after light irradiation (Figure 7b) [35,53]. Therefore, ·O_2_^−^ was proven to be produced and ·OH was absent during the light reaction, which was inconsistent with the result of radical quenching experiments.

The main degradation route of RhB using Bi_2_O_3_/BiOCl was deduced according to LC–MS analysis, as shown in Figure 7c. The probable degradation pathway was present in Figure 7d based on the detected intermediates in Figure 7c. Generally, RhB degradation included five main steps: N-de-ethylation, chromophore cleavage, de-carboxylation, ring opening, and mineralization [35,52]. RhB was decomposed by Bi_2_O_3_/BiOCl into various molecular fragments step by step under light irradiation gradually, and was completely degraded into CO_2_, H_2_O, and finally NH_4_^+^ [52,54].

Figure 8 illustrates the synthesis process of the Bi_2_O_3_/BiOCl nanosheet and its photocatalytic degradation mechanism of RhB. The as-prepared α-Bi_2_O_3_ was used as an adsorbent to remove Cl^−^ in acidic solution. The adsorbed Bi_2_O_3_ could transform the BiOCl and be further assembled into the heterostructured Bi_2_O_3_/BiOCl nanosheet [41]. Under visible light irradiation, the electrons on VB of Bi_2_O_3_/BiOCl were excited and transferred to CB of the material, leaving h^+^ on the VB. These photogenerated e^−^ can react with adsorbed O_2_ on the Bi_2_O_3_/BiOCl surface to produce abundant ·O_2_^−^ [55,56]. Such highly oxidizing ·O_2_^−^ and h^+^ can further react with RhB during the degradation process to generate small molecular substances (e.g., CO_2_ and H_2_O) [52,54].

Table 1 listed the photocatalytic performances of RhB for some recent representative catalysts in literature [57,58,59,60,61]. It could be found that our Bi_2_O_3_/BiOCl material exhibited a relatively short reaction time and high removal rate. Due to the distinctive structure and good photocatalytic activity, the α-Bi_2_O_3_/BiOCl nanosheet could be considered as a promising photocatalyst of organic dye.

## 4. Conclusions

The as-prepared α-Bi_2_O_3_ was used to remove Cl^−^ in acidic solutions to generate BiOCl. The α-Bi_2_O_3_/BiOCl heterostructured photocatalyst could be synthetized by coupling Bi_2_O_3_ with layered BiOCl. The composite material displayed the staggered nanosheet shape with a diameter of 80–200 nm. Both heterogeneous interface and nanosheet structure enhanced the photocatalytic activity of the material. For example, the Bi_2_O_3_/BiOCl binary catalyst delivered an excellent degradation rate of 99.9% for RhB at 60 min of visible light irradiation, which was much higher than 36.0% of Bi_2_O_3_ and 74.4% of BiOCl. Radical quenching experiments and ESR spectra confirmed that h^+^ and ·O_2_^−^ dominated the photocatalytic degradation process of RhB. The probable intermediates of RhB during the photodegradation process were further investigated using the LC–MS analysis. This work constructed a heterostructured Bi_2_O_3_/BiOCl photocatalyst with high photocatalytic performances, which provided a new opportunity for potential commercial applications.

## Figures and Tables

**Figure 1 nanomaterials-12-03631-f001:**
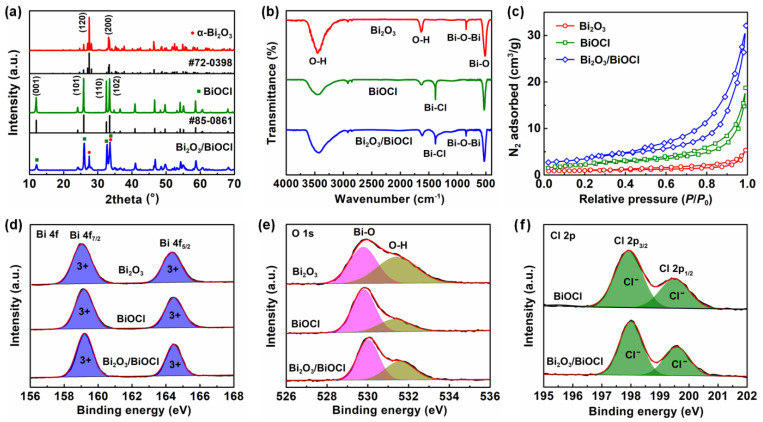
Physical properties of Bi_2_O_3_, BiOCl and Bi_2_O_3_/BiOCl: (**a**) XRD patterns, (**b**) FTIR spectra, (**c**) liquid nitrogen adsorption–desorption isotherms; high-resolution XPS spectra of (**d**) Bi 4f, (**e**) O 1s and (**f**) Cl 2p.

**Figure 2 nanomaterials-12-03631-f002:**
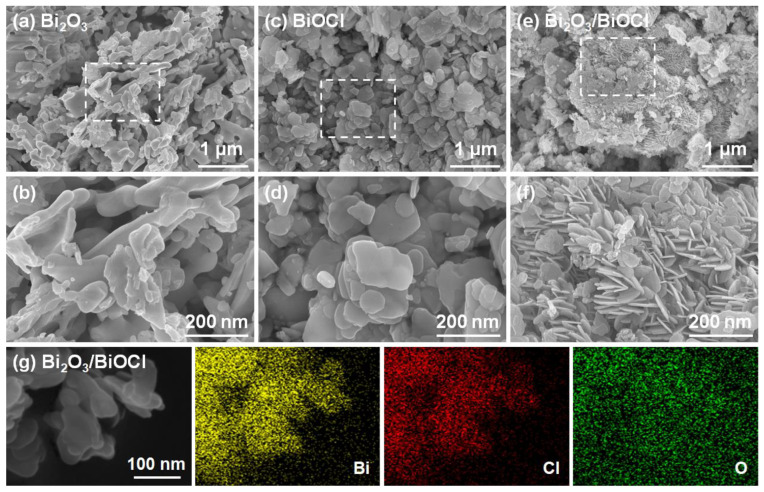
SEM images of (**a**,**b**) Bi_2_O_3_, (**c**,**d**) BiOCl and (**e**,**f**) Bi_2_O_3_/BiOCl; (**g**) mapping images of Bi, Ci and O elements for Bi_2_O_3_/BiOCl.

**Figure 3 nanomaterials-12-03631-f003:**
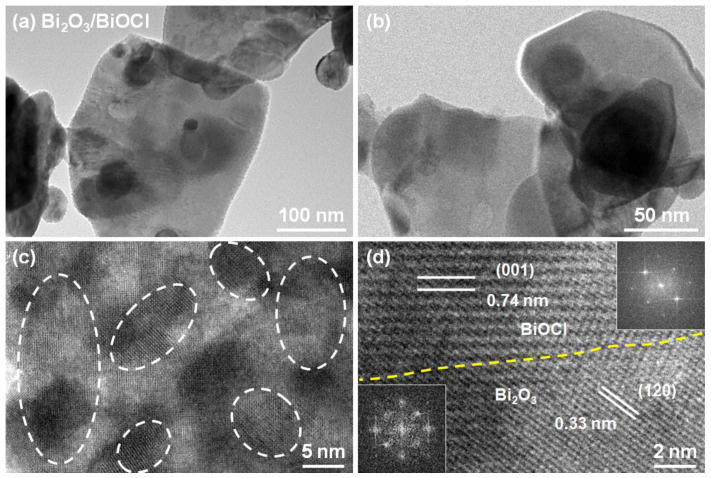
(**a**,**b**) TEM images of Bi_2_O_3_/BiOCl nanosheet; (**c**,**d**) HR-TEM images of Bi_2_O_3_/BiOCl with clear lattice fringes (inset in figure d was the FFT patterns).

**Figure 4 nanomaterials-12-03631-f004:**
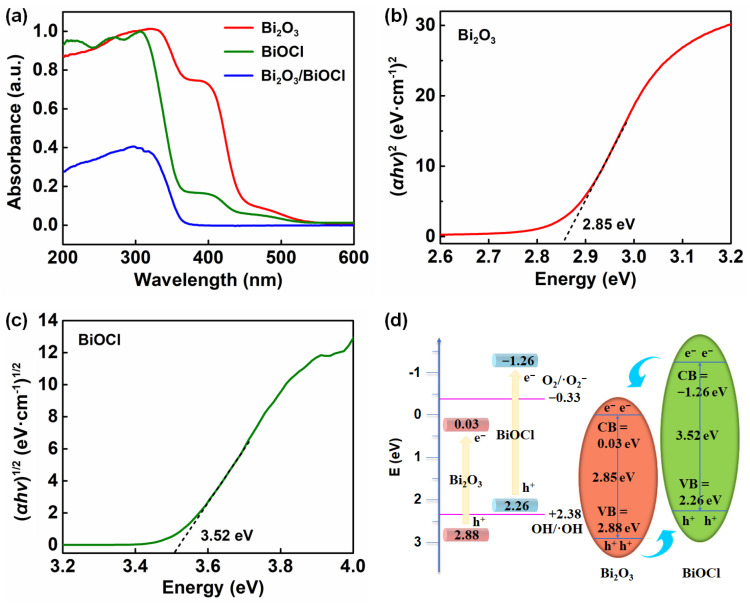
(**a**) UV–Vis DRS spectra of Bi_2_O_3_, BiOCl and Bi_2_O_3_/BiOCl; band gap energy *E*_g_ of (**b**) Bi_2_O_3_ and (**c**) BiOCl; (**d**) schematic illustration of band gap structure of Bi_2_O_3_ and BiOCl.

**Figure 5 nanomaterials-12-03631-f005:**
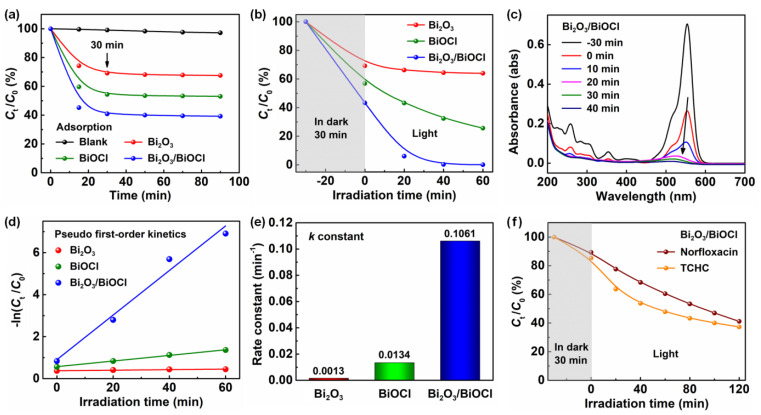
Photocatalytic degradation performances of RhB using Bi_2_O_3_, BiOCl and Bi_2_O_3_/BiOCl: (**a**) the adsorption curves of RhB onto three materials in darkness; (**b**) total photocatalytic degradation curves of RhB using three materials under light irradiation; (**c**) UV–Vis absorbance spectra of RhB using Bi_2_O_3_/BiOCl under different irradiation times; (**d**) linear fitting of pseudo first-order kinetic equation, (**e**) the histogram of apparent rate constant *k*; (**f**) total photocatalytic degradation curves of norfloxacin and TCHC using Bi_2_O_3_/BiOCl.

**Figure 6 nanomaterials-12-03631-f006:**
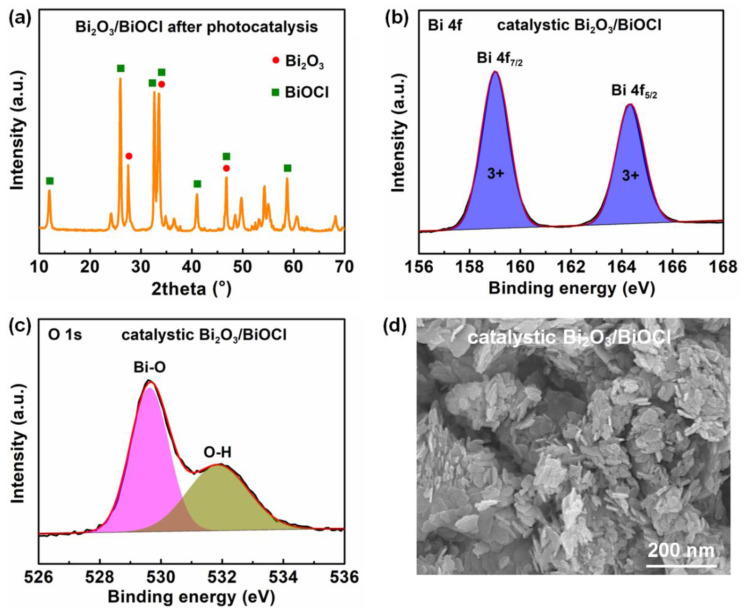
Stability evaluation for Bi_2_O_3_/BiOCl after 60 min photocatalytic degradation of RhB: (**a**) XRD pattern of sample; XPS spectrum of (**b**) Bi 4f and (**c**) O 1s; (**d**) SEM image.

**Figure 7 nanomaterials-12-03631-f007:**
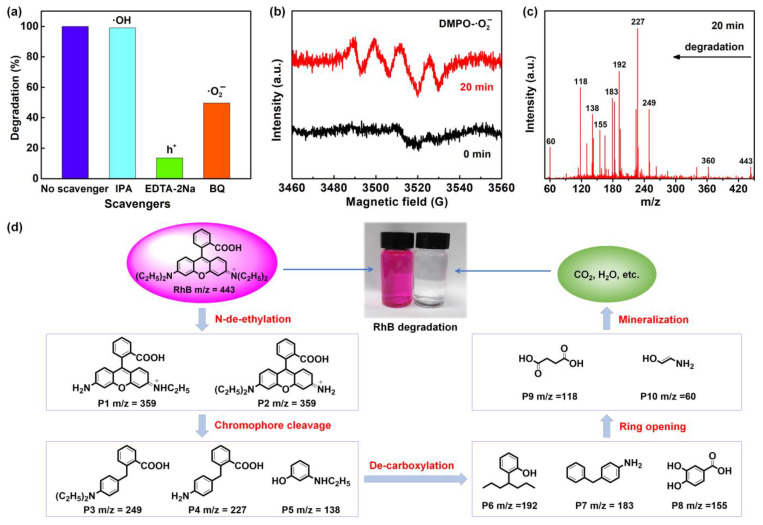
(**a**) Effects of different scavengers during the RhB photocatalytic degradation process; (**b**) ESR spectra by adding DMPO to capture ⋅O_2_^−^ with Bi_2_O_3_/BiOCl; (**c**) LC–MS (positive ESI scan) spectra of the photocatalytic degradation of RhB; (**d**) the probable degradation pathway of RhB by Bi_2_O_3_/BiOCl.

**Figure 8 nanomaterials-12-03631-f008:**
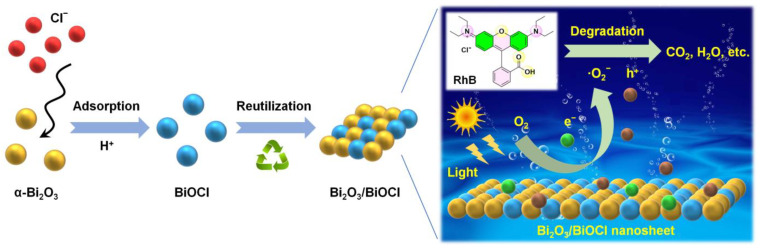
Schematic illustration of Bi_2_O_3_/BiOCl nanosheet and its photodegradation of RhB.

**Table 1 nanomaterials-12-03631-t001:** The photocatalytic degradation of RhB for some representative catalysts.

Photocatalyst	Dosage (mg/mL)	RhB Concentration (mg/L)	Light Source	Reaction Time (min)	Removal Rate (%)	Ref.
Bi_2_O_3_/BiOCl	20/100	20	300 W Xe lamp	60	99.9	This work
Bi_2_O_3_/Bi_2_S_3_	50/100	20	300 W Xe lamp	90	99.7	[57]
Ag_2_O/TiO_2_	130/100	4.8	visible-light	80	87.7	[10]
MoS_2_/NiFe	100/100	20	300 W Xe lamp	120	90	[58]
C_3_N_4_/ZnO	100/100	10	300 W Xe lamp	90	98.5	[59]
ZnO/Bi_2_MoO_6_	25/100	10	15 W cool daylight lamp	180	99.3	[60]
AgI/Bi_3_O_4_Br	20/100	50	300 W Xe lamp	60	98	[61]
Ti_3_C_2_/TiO_2_/BiOCl	100/100	10	500 W Xe lamp	120	84	[49]

## Data Availability

Not applicable.

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
