# Peer review of "Heterostructured α-Bi2O3/BiOCl Nanosheet for Photocatalytic Applications"

_nanomaterials, 2022, doi:10.3390/nano12203631_

Round 1

Reviewer 1 Report

Teng and coworkers have herein described an interesting piece of work where they have been capable not only to remove chlorides from acidic solutions but also to degrade Rhodamine B under visible light irradiation. For this purpose,  a heterostructured α-Bi2O3/BiOCl 2 nanosheet has been synthezied, fully characterized and its photoactivity has been investigated. The manuscript is well-writen and well-structured whereas the conclusions are consistent with the planned objectives. Therefore, from the point of view of this reviewer, this work can be published as is.   

Author Response

Dear Reviewer,

Thank you very much for what you have done during the reviewing process of our manuscript (nanomaterials-1965565) entitled “Heterostructured α-Bi2O3/BiOCl nanosheet for photocatalytic applications”. According to the comments, we carefully revised the manuscript and all the revisions were marked in red color in the revised manuscript.

Reviewer 1:

Teng and coworkers have herein described an interesting piece of work where they have been capable not only to remove chlorides from acidic solutions but also to degrade Rhodamine B under visible light irradiation. For this purpose, a heterostructured α-Bi2O3/BiOCl nanosheet has been synthezied, fully characterized and its photoactivity has been investigated. The manuscript is well-written and well-structured whereas the conclusions are consistent with the planned objectives. Therefore, from the point of view of this reviewer, this work can be published as is.

Response: We thank the reviewer for the positive comments on our manuscript.

Thanks again for taking time to read our manuscript! We really appreciate your suggestions.

Sincerely yours,

A/Prof. Ying Zhang

School of Chemical Engineering & Technology and Zhongyuan Critical Metals Laboratory

Zhengzhou University

Zhengzhou, Henan 450001, P.R. China

[email protected]

Reviewer 2 Report

This research paper entitled “Dechlorination product of heterostructured α-Bi2O3/BiOCl nanosheet towards enhanced photocatalyst for organics degradation” demonstrated an excellent photocatalytic activity of the heterostructured α-Bi2O3/BiOCl nanosheets for Rhodamine B (RhB) degradation under visible light.

The paper is well organized and easy to read. All characterization methods were appropriately presented . As a fundamental research work on nanomaterials, this paper can be accepted to be published in Nanomaterials after minor revision.

     1)      The last sentence of Abstract :   

     2)      For DRX indexation, please precise the database of PDF# (JCPDS or COD or ICDD)

     3)      Lines189-200: bold characters are not necessary.

Author Response

Dear Reviewer,

Thank you very much for what you have done during the reviewing process of our manuscript (nanomaterials-1965565) entitled “Heterostructured α-Bi2O3/BiOCl nanosheet for photocatalytic applications”. According to the comments, we carefully revised the manuscript and all the revisions were marked in red color in the revised manuscript. On behave of all authors, I would like to answer the reviewer’s comments point by point.

Reviewer 2:

This research paper entitled “Dechlorination product of heterostructured α-Bi2O3/BiOCl nanosheet towards enhanced photocatalyst for organics degradation” demonstrated an excellent photocatalytic activity of the heterostructured α-Bi2O3/BiOCl nanosheets for Rhodamine B (RhB) degradation under visible light.

The paper is well organized and easy to read. All characterization methods were appropriately presented. As a fundamental research work on nanomaterials, this paper can be accepted to be published in Nanomaterials after minor revision.

Response: We thank the reviewer for the positive comments on our manuscript.

Comment 1: The last sentence of Abstract.

Response: Thanks for your comment! The last sentence of Abstract was revised as “This work develops a green strategy to synthesize high-performance photocatalyst for organic dye degradation.”

Comment 2: For XRD indexation, please precise the database of PDF# (JCPDS or COD or ICDD).

Response: We accept your helpful advice. The selected database of PDF# was JCPDS.

Comment 3: Lines 189-200: bold characters are not necessary.

Response: Thanks for your reminding! The bold characters in above region (TEM) were revised.

Thanks again for taking time to read our manuscript! We really appreciate your suggestions.

Sincerely yours,

A/Prof. Ying Zhang

School of Chemical Engineering & Technology and Zhongyuan Critical Metals Laboratory

Zhengzhou University

Zhengzhou, Henan 450001, P.R. China

[email protected]

Reviewer 3 Report

(1) Introduction line 48: the performance of b-Bi2O3 in combination with BiOCl showed only 80% within 2.5 hrs, and the result of this work is much better (99.9 % in 40 mins). However, the band gap of b-Bi2O3 is 2.4 eV which is smaller than a-Bi2O3 (2.8 eV). Therefore, in this context, the authors need to articulate the reasons for such differences in detail. Also, please, see this: https://doi.org/10.1007/s11244-022-01644-z.

(2) In SEM, what is the reason that chlorination changed the shape from irregular nanorods to nanoflakes?

(3) In fig 6c, O 1s peak of Bi-O-Bi is said to be maintained with pristine composite; however, fig shows that there is a shift in value what is the reason for shifting the value?

(4) In HRXRD, as per PDF #72-0398 there is no mention of the peak at 33.3 corresponding with 200 crystal planes of a-Bi2O3. Also, in reference 31 there is no PDF #85-0861 mentioned. In summary, the references should be reviewed minutely in the place they are cited.

(5) Table 1 lacks a column that indicates the source of the light, which will help to make the comparison easy and clear. Therefore, it is recommended to include the missing part.

(6) This manuscript does not speak well about adsorption experimentation, for instance, conditions and parameters, because as per fig 5a adsorption data has been shown for 90 mins. For reproducibility, detail is required. Is there any reason for not mentioning this in detail? Also, in line (111) some detail should be provided about the lamp, for instance, its specification.  

(7) This manuscript has shown good results; however, this work seems not to be completely new. Therefore, the manuscript should clearly demonstrate the novelty and motivation in the introduction part. Also, follow this paper (https://doi.org/10.1016/j.jphotochem.2022.114066).

(8) English language should be reviewed once minutely. For example, it seems typos mistakes in line 135 “plans.”

Once the above queries are addressed, this work would be a good scientific contribution to science and society.

Author Response

Dear Reviewer,

Thank you very much for what you have done during the reviewing process of our manuscript (nanomaterials-1965565) entitled “Heterostructured α-Bi2O3/BiOCl nanosheet for photocatalytic applications”. According to the comments, we carefully revised the manuscript and all the revisions were marked in red color in the revised manuscript. On behave of all authors, I would like to answer the reviewer’s comments point by point.

Reviewer 3:

Comment 1: Introduction line 48: the performance of b-Bi2O3 in combination with BiOCl showed only 80% within 2.5 hrs, and the result of this work is much better (99.9 % in 40 mins). However, the band gap of b-Bi2O3 is 2.4 eV which is smaller than a-Bi2O3 (2.8 eV). Therefore, in this context, the authors need to articulate the reasons for such differences in detail. Also, please, see this: https://doi.org/10.1007/s11244-022-01644-z.

Response: We appreciate your suggestion. This paper was added as new reference [32]. According to this reference, the heterojunctions possess an enhanced photocatalytic activity compared to pure Bi2O3 and BiOCl due to efficient charge separation and transfer across the interface. Accordingly, our heterostructured α-Bi2O3/BiOCl delivered higher degradation rate than b-Bi2O3 in previous report. Please check the updated description in Introduction. (Page 2, Line 70)

Comment 2: In SEM, what is the reason that chlorination changed the shape from irregular nanorods to nanoflakes?

Response: We thank the reviewer for pointing this out. The BiOCl has intrinsic layered structure [22,24]. After chlorination, the Bi2O3 nanorods converted to BiOCl nanoflakes.

Comment 3: In fig 6c, O 1s peak of Bi-O-Bi is said to be maintained with pristine composite; however, fig shows that there is a shift in value what is the reason for shifting the value?

Response: We thank the reviewer for raising this question. The Bi-O-Bi deconvolutional peaks of pristine and catalytic Bi2O3/BiOCl located at 529.9 eV (Fig. 1e) and 529.6 eV (Fig. 6c), respectively. A slight shift toward lower binding energy (~0.3 eV) of Bi-O peak was ascribed to the lower oxidation state of Bi atoms [43]. Please check the updated description. (Page 9, Line 280-281)

Comment 4: In HRXRD, as per PDF #72-0398 there is no mention of the peak at 33.3 corresponding with 200 crystal planes of a-Bi2O3. Also, in reference 31 there is no PDF #85-0861 mentioned. In summary, the references should be reviewed minutely in the place they are cited.

Response: Thanks for your carefulness! After careful inspection, the reference is correct.

Comment 5: Table 1 lacks a column that indicates the source of the light, which will help to make the comparison easy and clear. Therefore, it is recommended to include the missing part.

Response: We accept your helpful suggestion. The information of light source has added in Table 1.

Table 1. The photocatalytic degradation of RhB for some representative catalysts.

Photocatalyst

Dosage (mg/mL)

RhB concentration (mg/L)

Light source

Reaction time (min)

Removal rate (%)

Ref.

Bi2O3/BiOCl

20/100

20

300 W Xe lamp

60

99.9

This work

Bi2O3/Bi2S3

50/100

20

300 W Xe lamp

90

99.7

[58]

Ag2O/TiO2

130/100

4.8

visible-light

80

87.7

[10]

MoS2/NiFe

100/100

20

300 W Xe lamp

120

90

[59]

C3N4/ZnO

100/100

10

300 W Xe lamp

90

98.5

[60]

ZnO/Bi2MoO6

25/100

10

15 W cool daylight lamp

180

99.3

[61]

AgI/Bi3O4Br

20/100

50

300 W Xe lamp

60

98

[62]

Ti3C2/TiO2/BiOCl

100/100

10

500 W Xe lamp

120

84

[49]

Comment 6: This manuscript does not speak well about adsorption experimentation, for instance, conditions and parameters, because as per fig 5a adsorption data has been shown for 90 mins. For reproducibility, detail is required. Is there any reason for not mentioning this in detail? Also, in line (111) some detail should be provided about the lamp, for instance, its specification.

Response: We appreciate your advice. The adsorption experiments were conducted as the same as photocatalytic degradation, just without light radiation during whole process: 20 mg Bi2O3 or Bi2O3/BiOCl was added into 100 mL RhB solution (20 mg/L) under stirring of 300 RPM at room temperature in dark. The specification of used lamp was 300 W Xe lamp (Asahi Spectra, MAX-303). Please check the update information. (Page 3, Line 114-117)

Comment 7: This manuscript has shown good results; however, this work seems not to be completely new. Therefore, the manuscript should clearly demonstrate the novelty and motivation in the introduction part. Also, follow this paper (https://doi.org/10.1016/j.jphotochem.2022.114066).

Response: Thanks for the insightful comment. This paper was added as new reference [23]. The novelty of this manuscript could be summarized as: (1) Previous studies mainly focused on β-Bi2O3 (thermodynamic metastable phase), and there were a few reports about stable α-Bi2O3; (2) Traditionally, BiOCl is prepared through the reaction between Bi2O3 and HCl, while our BiOCl was transformed from Bi2O3 after removing Cl in acidic wastewater; (3) Bi2O3/BiOCl can serve as “heterojunction” nanosheet to modify electronic band structure and enhance the photocatalytic activity due to efficient charge separation and transfer across the interface. Please check the updated description. (Page 2, Line 45-53, Line 68-70)

Comment 8: English language should be reviewed once minutely. For example, it seems typos mistakes in line 135 “plans”.

Response: Thanks for your reminding. The word “plans” was revised as “planes”. Also, the English in manuscript was polished.

Thanks again for taking time to read our manuscript! We really appreciate your suggestions.

Sincerely yours,

A/Prof. Ying Zhang

School of Chemical Engineering & Technology and Zhongyuan Critical Metals Laboratory

Zhengzhou University

Zhengzhou, Henan 450001, P.R. China

[email protected]

Reviewer 4 Report

This manuscript is dedicated to a new synthesis scheme of heterostructured α-Bi2O3/BiOCl nanosheet with potential applications as enhanced photocatalyst for organics degradation. Not only the proposed preparation scheme is an interesting approach both practical and and reliable but such type of schemes have not been studied well yet. The characterization efforts are very well systematized and convincing. The discussion provided is quite adequate for the present purpose. The well detailed and at the same time comparative context of the results clarifies convincingly the researched experimental scheme and prompts a good understanding how to synthesize high-performance and low-cost photocatalyst at the basis of the heterostructured α-Bi2O3/BiOCl nanosheet which can benefit wide range of developing research and applications.

From practical point of view, the reported results thus bring new knowledge and certainly represent an original contribution in the present context.

The authors chose an adequate structure of the manuscript – an excellent point of departure for such a study. Also, they provided a balanced realistic and nicely illustrated presentation of their results and corresponding analysis that is of much scientific and practical interest and adds new knowledge to the field.

The present manuscript is a significant contribution, this work once published would be instructive and suggestive in terms of further studies and to a wider readership.

There are some minor issues with this already excellent manuscript that will need to be addressed before becoming suitable for publication, i.e., it can be considered for publication after a minor revision:

1: Title is a little bit heavy, imprecise, and not attractive to wider audience, it should be shortened and made more focused; I would suggest something like “Heterostructured α-Bi2O3/BiOCl nanosheet for photocatalytic applications”

2: Mentioning TiO2 in the abstract is inappropriate since it is not any direct subject of the research described and reported in this manuscript.

3: In the introduction, the authors miss that a wide range of theoretical/modelling/simulation works have already been adopted/used for studying the synthesis of similar heterostructures as the one used and employed in the present work (some of them even including Bi, as well as other elements with similar chemical properties, as well as oxides). Examples in which such theoretical works help understanding synergies and synthesis issues, including chemistry of synthesis, and even directly guide experimental work include Journal of Physics: Condensed Matter 27 (2015) 485306 and Applied Surface Science 548 (2021) 149275. Such works should be referenced for achieving a clear picture of viability of heterostructures-based model systems for the purposes of the present work.

4: Authors should mention, elaborate and be more specific about any concrete evaluation of the thermal stability range applicable to the (as synthesized) heterostructured α-Bi2O3/BiOCl nanosheet.

5: Spell-check and stylistic revision of the paper are still necessary. Some, long sentences, misspellings, etc., still are noticeable throughout the text.

Author Response

Dear Reviewer,

Thank you very much for what you have done during the reviewing process of our manuscript (nanomaterials-1965565) entitled “Heterostructured α-Bi2O3/BiOCl nanosheet for photocatalytic applications”. According to the comments, we carefully revised the manuscript and all the revisions were marked in red color in the revised manuscript. On behave of all authors, I would like to answer the reviewer’s comments point by point.

Reviewer 4:

This manuscript is dedicated to a new synthesis scheme of heterostructured α-Bi2O3/BiOCl nanosheet with potential applications as enhanced photocatalyst for organics degradation. Not only the proposed preparation scheme is an interesting approach both practical and and reliable but such type of schemes have not been studied well yet. The characterization efforts are very well systematized and convincing. The discussion provided is quite adequate for the present purpose. The well detailed and at the same time comparative context of the results clarifies convincingly the researched experimental scheme and prompts a good understanding how to synthesize high-performance and low-cost photocatalyst at the basis of the heterostructured α-Bi2O3/BiOCl nanosheet which can benefit wide range of developing research and applications.

From practical point of view, the reported results thus bring new knowledge and certainly represent an original contribution in the present context.

The authors chose an adequate structure of the manuscript – an excellent point of departure for such a study. Also, they provided a balanced realistic and nicely illustrated presentation of their results and corresponding analysis that is of much scientific and practical interest and adds new knowledge to the field.

The present manuscript is a significant contribution, this work once published would be instructive and suggestive in terms of further studies and to a wider readership.

There are some minor issues with this already excellent manuscript that will need to be addressed before becoming suitable for publication, i.e., it can be considered for publication after a minor revision.

Response: We thank the reviewer for the positive comments on our manuscript.

Comment 1: Title is a little bit heavy, imprecise, and not attractive to wider audience, it should be shortened and made more focused; I would suggest something like “Heterostructured α-Bi2O3/BiOCl nanosheet for photocatalytic applications”.

Response: We accept your helpful advice. The title was revised as “Heterostructured α-Bi2O3/BiOCl nanosheet for photocatalytic applications” in the manuscript revision.

Comment 2: Mentioning TiO2 in the abstract is inappropriate since it is not any direct subject of the research described and reported in this manuscript.

Response: We accept your suggestion. The mentioned TiO2 was deleted and description was revised.

Comment 3: In the introduction, the authors miss that a wide range of theoretical/modelling/ simulation works have already been adopted/used for studying the synthesis of similar heterostructures as the one used and employed in the present work (some of them even including Bi, as well as other elements with similar chemical properties, as well as oxides). Examples in which such theoretical works help understanding synergies and synthesis issues, including chemistry of synthesis, and even directly guide experimental work include Journal of Physics: Condensed Matter 27 (2015) 485306 and Applied Surface Science 548 (2021) 149275. Such works should be referenced for achieving a clear picture of viability of heterostructures-based model systems for the purposes of the present work.

Response: We thank the reviewer for pointing this out. These two papers have added as new references [22,25]. Also, the related description of heterostructure was also added in Introduction. (Page 2, Line 49-53)

Comment 4: Authors should mention, elaborate and be more specific about any concrete evaluation of the thermal stability range applicable to the (as synthesized) heterostructured α-Bi2O3/BiOCl nanosheet.

Response: Thanks for the insightful comment. The TG (25−900 °C) analysis was used to investigate the thermal stability of Bi2O3/BiOCl. From Figure S3, the small weight loss on TG curve below 700 °C could be assigned to the absorbed water. Until the temperature rose to 700 °C, an obvious decline started to appear at 700−800 °C, attributing to pyrolysis of BiOCl composition. The result revealed the excellent thermal stability of such the Bi2O3/BiOCl material. Please check the updated Figure S3 (SI file) and description. (Page 4, Line 167-171)

New Figure S3. The TG curve of Bi2O3/BiOCl in nitrogen atmosphere.

Comment 5: Spell-check and stylistic revision of the paper are still necessary. Some, long sentences, misspellings, etc., still are noticeable throughout the text.

Response: Thanks for your reminding. The English was polished.

Thanks again for taking time to read our manuscript! We really appreciate your suggestions.

Sincerely yours,

A/Prof. Ying Zhang

School of Chemical Engineering & Technology and Zhongyuan Critical Metals Laboratory

Zhengzhou University

Zhengzhou, Henan 450001, P.R. China

[email protected]

Round 2

Reviewer 3 Report

The revision is satisfactory. I would recommend its publication as is.